# Prediction of Lupus Classification Criteria via Generative AI Medical Record Profiling

**DOI:** 10.3390/biotech14010015

**Published:** 2025-03-06

**Authors:** Sandeep Nair, Gerald H. Lushington, Mohan Purushothaman, Bernard Rubin, Eldon Jupe, Santosh Gattam

**Affiliations:** Progentec Diagnostics, Inc., 755 Research Pkwy, Oklahoma City, OK 73104, USA; sandeepnair@progentec.com (S.N.); mpurushothaman@progentec.com (M.P.); brubin@progentec.com (B.R.); ejupe@progentec.com (E.J.); sgattam@progentec.com (S.G.)

**Keywords:** systemic lupus erythematosus (SLE), medical records (MRs), large language model (LLM), natural language processing (NLP), generative artificial intelligence (genAI), American College of Rheumatology (ACR)

## Abstract

Systemic lupus erythematosus (SLE) is a complex autoimmune disease that poses serious long-term patient burdens. **(1)** Background: SLE patient classification and care are often complicated by case heterogeneity (diverse variations in symptoms and severity). Large language models (LLMs) and generative artificial intelligence (genAI) may mitigate this challenge by profiling medical records to assess key medical criteria. **(2)** Methods: To demonstrate genAI-based profiling, ACR (American College of Rheumatology) 1997 SLE classification criteria were used to define medically relevant LLM prompts. Records from 78 previously studied patients (45 classified as having SLE; 33 indeterminate or negative) were computationally profiled, via five genAI replicate runs. **(3)** Results: GenAI determinations of the “Discoid Rash” and “Pleuritis or Pericarditis” classification criteria yielded perfect concurrence with clinical classification, while some factors such as “Immunologic Disorder” (56% accuracy) were statistically unreliable. Compared to clinical classification, our genAI approach achieved a 72% predictive success rate. **(4)** Conclusions: GenAI classifications may prove sufficiently predictive to aid medical professionals in evaluating SLE patients and structuring care strategies. For individual criteria, accuracy seems to correlate inversely with complexities in clinical determination, implying that improvements in AI patient profiling tools may emerge from continued advances in clinical classification efficacy.

## 1. Background

Characterized by long durations of care and presenting substantial symptom heterogeneity, many autoimmune conditions pose unique medical challenges. Notable among these challenging disorders is systemic lupus erythematosus (SLE), an uncurable inflammatory condition presenting multisystem symptom effects (including in skin, joints, mucocutaneous tissue and blood and other detectable effects) and a potentially serious risk of organ damage (especially kidneys, lungs, heart, central nervous system, etc.) [1], affecting approximately 72.8 people per 100,000 in the United States [2]. In terms of healthcare costs, the burden of treating autoimmune patients is significant due to the long duration and high levels of patient–physician interaction. According to a review by Kariburyo et al., the average patient averages roughly 26 physician visits during the first year after SLE classification, plus roughly two outpatient and two ER admissions [3]. Concomitantly, this same review estimated SLE-related patient medical costs of roughly USD 8712 per year (unadjusted estimates from 2015) in the United States, leading to a national medical burden of between USD 1.4B and USD 3.2B (estimates from 2013 to 2015). Beyond such direct costs, broader impacts on patients and society are substantial [4], as typified by elevated levels of unemployment or work-related absenteeism among SLE patients [5], as well as elevated risks of morbidity and mortality [6,7].

Although statistics from the Kariburyo review show a significant increase in relative medical resource utilization during the first year after initial SLE classification, early classification tends to produce substantially better patient outcomes [8,9,10,11]. The definitive classification/diagnosis of autoimmune disorders like SLE is challenging. Complicated by frequent misdiagnosis [12], recent estimates for the time to reach SLE disease classification after initial symptoms have ranged from 2 to 6 years [13,14], with slower classification typically producing tangibly negative health repercussions [14]. Patients with SLE require life-long treatment from the time of diagnosis and risk intermittent flares that can impose major limitations on patient functioning and threaten irreversible organ damage [7]. Effective medical treatments may significantly reduce these impacts, but such efforts are hindered by challenges associated with differentiating SLE patients from those suffering other connective tissue disorders, monitoring disparate symptoms and complications among patients with known SLE and mapping patient-specific paths toward case stabilization or remission [8]. These burdens collectively motivate the search for new resources to improve disease classification and care, via reduced costs and enhanced outcomes.

One avenue for potential improvement entails developing intelligent technology to assist medical professionals in the classification of SLE. Both initial classification [9,10,11] and the long-term monitoring of evolving disease activity [13,14,15,16,17,18,19,20,21,22,23] are informationally dense challenges that may be met in part through the development and validation of a patient profiler tool for systematic assimilation and encapsulation of medical annotations and laboratory test results from medical records. A useful patient profiler could thus foster information management to address disparate symptoms and indications intended to inform initial SLE classification of patients, and to evaluate the wide range of comparable medical measures available for distinguishing between patients with stable cases, versus suggesting current or emerging problems requiring adaptive care strategies.

To achieve this, a patient profiler would require target foci in the form of symptoms or observables to track, as specified according to relevant medical priorities. For SLE, this may include constitutional, neuropsychiatric, mucocutaneous and musculoskeletal conditions assessed by standard clinical examinations, plus quantitative measurements from blood and serum tests, such as dsDNA, and anti-Smith antibodies that target common core proteins of U1, U2, U4 and U5 snRNP [9]. Various other relevant measures may also include those determined from medical imaging or biopsy.

Taken as individual criteria, the list of relevant measures is complex and would be difficult to meaningfully assess. Rather, a pragmatic way to subdivide the list and meaningfully assess criterial performance is to systematically examine target metrics within the larger framework of an existing clinical standard for evaluating patients, potentially in terms of symptoms or metrics suggestive of hitherto unclassified SLE [9,10,11] or, in the case of patients with known SLE, measures potentially indicative of disease state [22,23,24]. From these various standards, we have chosen to target our pilot study toward reproducing SLE classification criteria specified within a widely used American College of Rheumatology (ACR) index [9], hereafter referred to as “ACR 1997”.

The ACR 1997 protocol blends descriptive observations and quantitative qualifiers whose automated co-assimilation may currently be best addressed through specialized artificial intelligence (A.I.) algorithms. For such tasks, conventional machine learning algorithms may not prove ideal, since these algorithms are more tailored for quantitative (i.e., lab-based) diagnostic prediction [24,25], with less proven efficacy toward natural language processing (NLP). Similarly, convolutional neural networks have proven adept at imaging-based determinations [26] but are not optimized for contextualizing text documents. By contrast, a new promising strategy, known as generative A.I. (genAI), for processing mixed descriptive/quantitative textual information entails using large language models (LLMs) for intelligently parsing medical records, followed by the application of generative adversarial networks to reassemble extracted information according to specified formats (e.g., reports, summaries or answers to query responses) [27,28,29].

In operative practice, genAI proceeds from an intelligent query (often phrased in terms of human language), to draw information (in reality, word patterns) from an LLM which encodes representative language trends present in an extensive body of literature. For a medically relevant LLM, this may include case studies, protocols, a broad sampling of anonymized medical records, scientific papers, etc. Given comparable instances within the LLM, genAI then proceeds to predict what might be a “reasonable” (ideally logical and fact-based) response to the original query. For illustrative purposes, a successful query could be formulated along the lines of something as humanly intuitive as follows:



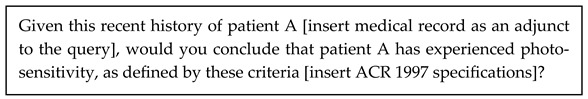



To illustrate the current technical aptitude of genAI, studies from the past year found multiple LLMs with research analytical capacities commensurate with competent (i.e., third quartile) students pursuing advanced degrees (M.D. or post-graduate) [30]. Simulated genAI communication attained skills sufficient to assist in practical instruction, either as teaching assistants or simulated patients [31]. More to the immediate point of this article, a meta-analysis of published research between 2020 and mid-2023 suggested a practical value of genAI in healthcare areas such as medical annotation, reporting, disease classification and screening [32].

Given such evidence of practical value, this manuscript explores the development and demonstration of a genAI-based patient-specific case profiler, trained to report specific medical observables based on real-world medical data.

Given suitable algorithms and source data, the final prerequisite for developing an automated patient profiler is to set reasonable specifications. In general, the profiler would thus be expected to deliver (i) timely reporting, as addressed by computational automation, i.e., the adoption of IT protocols for fast, automatic transmission of pre-specified types of information from pre-specified sources. The second and third specifications then require that such information be both (ii) succinct and (iii) accurate. These stipulations require contextual “awareness” of the sort once reserved for human intelligence, but now increasingly emulated using AI. Currently, one of the best demonstrated strengths of genAI is its ability to tolerate case heterogeneity while reliably distilling complex information into simple intuitive statements [33,34], thus potentially addressing the requirement for succinctness. Assessing the final requirement for accuracy forms the primary basis of this paper, wherein an objective attempt is made to assess how well current, representative genAI methods can parse medical records and determine relevant observables or classification criteria with passable accuracy relative to controls derived from instances where rheumatologists have already attempted such assessments.

## 2. Methods

Prerequisites for developing and validating a genAI tool for profiling patients according to their medical records include (1) representative examples of medical observables to profile, (2) a reasonable sample of patient records from which to assess such observables, (3) a secure platform for acquiring and processing such information and (4) a protocol for implementing NLP and genAI models.

### 2.1. Medical Data

Data for this analysis were derived from medical records (at least one full year’s worth, in all cases) corresponding to 78 participants drawn from two of our clinical studies. Of these, 39 individuals (hereafter called “pre-SLE-classification” or “pre-SC”) suffered symptoms that resembled aspects of SLE, but had not, throughout the duration of the medical records, been classified as having SLE, prior to undergoing detailed clinical SLE classification at the study’s end [35]. An additional 39 (the “post-SLE-classification” or “post-SC”) cases were drawn at random from a different study whose patients had been classified as SLE-positive prior to the study [36]. In all cases, medical records, lab tests, patientreported outcomes and electronically tracked biometrics were made available for the genAI patient profiler development. Documentation from pre-SC cases also included clinical end-of-study SLE classification reports; these were not used to train our profiler, but did enable the rigorous validation of genAI-based criterial predictions. For all pre-SC and post-SC cases, medical test results were also provided for a variety of indications, including those relevant to SLE characterization (e.g., antinuclear antibodies and complement tests), plus metrics addressing various other analytically important targets.

### 2.2. Medical Observables

In terms of a complementary set of medical observables that one would expect to find represented within these medical records, the natural choice would be one of the various SLE patient evaluation standards [9,10,11,20,21,23]. For demonstration purposes, the choice can be determined by a balance of practicability and utility. Thus, the ACR 1997 set of SLE classification criteria was selected due to its compact nature (eleven independent indications; see Table 1) and the balance between qualitative information (e.g., text annotations in medical records) and quantitative test results (tabular data from laboratory test services). A patient that meets any four of the eleven criteria, serially or simultaneously, is classified as having SLE.

Results from the pre-SC patients provide example medical records for demonstrating genAI-based extraction and formulation of these criteria in possible SLE patients, plus two opportunities for validation: detailed criterion-level assessments, plus expert clinical determinations of SLE classification based on satisfaction either of the ACR criteria presented in Table 1, or analogous criteria from the newer ACR/EULAR standard [10]. Data from post-SC patients, by contrast, lacks clinical SLE determination documents, but provides comparable medical records and lab test results, and fosters a broad mechanism for validation in that all participants are presumed to be SLE-positive (i.e., as per ACR 1997 standards, all would be expected to manifest at least four positive criteria [9]).

### 2.3. Development Framework

The scheme for data acquisition and information processing is outlined in the upper left portion of Figure 1. Original inputs for patients were acquired from various text-based documents. To administer clinical studies, a HIPAA-compliant architecture was implemented, using a secure data repository from monday.com, Ltd. (a cloud-based work operating system), which is readily interfaceable with an Amazon Web Services S3 data bucket via secure connections enabled by Amazon Lambda functions. Files in the bucket were then available for contextual analyses, beginning with optical character recognition (OCR) via Amazon Textract (version 2022H2; a high-performance tesseract-based implementation). This process is largely automated, except for unusually large PDF files (e.g., the largest medical history we examined encompassed 887 pages), for which chunking was performed to ensure compliance with Textract asynchronous memory limits; i.e., long files were decomposed into 50-page increments, each of which was then processed separately, Finally, all increments from a given record file were later reconstituted as a single text file. Tabular data within the PDFs were extracted separately using Textract’s “Tables” feature.

### 2.4. Natural Language Processing and Generative A.I.

As shown near the center of Figure 1, NLP on the digitized text files was performed on Amazon Bedrock using the Claude-3-Haiku LLM, which has been optimized for a balance between high speed and accuracy [51]. The subsequent processing relies on detailed descriptive prompts to elicit desired pieces of information from the OCR-produced, patient-specific medical text (e.g., the presence or absence of a given laboratory test or physician’s annotation), which requires carefully chosen textual instructions (called “prompts”) that were developed in this study by extending the prompts to include representative examples of required output style and clear statements of the ACR 1997 criterial specifications, using significant medical detail—a strategy known as “cache-augmented generation” (CAG), as opposed to “retrieval-augmented generation” (RAG), where prompt contextuality is extended via external sources.

Accurate NLP requires prompt optimization based on domain experience (i.e., published reports and inputs from medical specialists) plus trial and error. Even for detailed prompts, genAI produces varying results which may be controlled by an algorithmic “temperature”, which is a measure of randomness such that queries with high values are more free to consider text of plausible relevance to the prompt, but are also more vulnerable to perceiving invalid relationships (a.k.a., hallucinations); those with low temperatures adhere more closely to the prompt, thus reducing errors of commission while increasing possible omissions. For this study, preliminary variation experiments led us to choose a conservative value of t = 0.1, in pair with a token size of 2048 (the text-window size used to determine context). The available “top_k” and “top_p” parameters were left at the default values for Claude-3 (5 and 0.7, respectively). To assess model stability and the statistical significance of specific findings, all documents were analyzed in five-fold replication. Predicted values (1 for apparent presence of the criterion; 0 for its absence) were then reported on a per-patient basis for all ACR component criteria. Concordances (correct/incorrect prediction relative to rheumatologists’ assessment) were determined for all cases where criterion-level rheumatologists’ assessments were available. Component-wise results were then recorded as comma-separated-value files in the S3 bucket for analysis and scoring.

Patient-specific five-fold-sampled ACR assessments form the basis for a standard, conveniently updateable report that can provide medical specialists with insight into patient health status, providing evidence regarding SLE-associated symptoms that could warrant more detailed examination and testing. To assess the significance of these determinations, consistency tests were implemented. To estimate case-specific consistency, full five-fold agreement (i.e., all five tests yielding a positive value for the criterion, or all five yielding a null result) was deemed to reflect strong consistency, a 4-1 split was considered moderately consistent, and a 3-2 split suggested lower confidence. To interpret these arbitrary labels, performance statistics were compiled to determine relative rates for each confidence level (i.e., consistent vs. moderately consistent vs. low confidence) for each ACR criterion.

Internal consistency must still be bolstered by confidence that genAI assessment can be reliably corroborated by clinical determinations. Thus, predictive accuracy was first estimated herein by statistical evaluation of criterion-level concordances relative to all recorded physician determinations for ACR criteria. Secondly, our profiler was assessed in a global manner by the summation of ACR criteria for patients, followed by evaluation of the concordance of ACR criteria sums relative to the known SLE classification status of all patients. Specifically, for a clinically known SLE-positive patient, a sum of predicted ACR criteria ≥ 4 reflected predictive agreement, while a sum < 4 constituted an error. Likewise, clinical assessments that failed to justify patient classification as SLE-positive would require a genAI-predicted ACR criterial sum < 4 to infer concordance.

## 3. Results

The analytical reliability of our profiler determinations was evaluated according to their capacity to reproduce clinical determinations for individual ACR 1997 criteria, with further validation according to concordance between genAI-based SLE classifications and clinically determined patient classifications.

### 3.1. Criterion-Level Predictions

Table 2 summarizes genAI predictive confidence for the eleven ACR criteria, across a set of 27 patients from the pre-SC set for whom full clinical criterial determinations were made. The rightmost column determines overall concordance between clinical determinations and genAI. Performance ranged from two criteria (“2. Discoid Rash” and “6. Pleuritis or Pericarditis”) for which full agreement between genAI and clinical determinations was observed across all patients, down to three criteria (“4. Oral Ulcers”, “5. Nonerosive Arthritis” and “8. Immunologic Disorder”) for which predictive success did not achieve statistical significance (*p* < 0.05). A more detailed decomposition of external criterial predictivity statistics is provided in Appendix A.

The central columns assess internal consistency (i.e., whether the algorithm consistently reaches the same conclusion) in the same set of input medical records. Over five replicate samplings, internal stability ranged from a perfect score for assessing “Discoid Rash” (i.e., all five replicates yielded the same results for all 78 cases) down to a low value for “Immunologic Disorder”, for which only 74% of patients yielded perfect 5-0 consistency in predictions, while 17% of patients had a single dissenting replicate (i.e., four replicates agreed, one disagreed) and 9% of cases yielded an ambiguous 3-2 split.

External predictivities (i.e., the rate at which genAI-predicted criteria agreed with clinically determined criteria) for “Discoid Rash” and “Immunologic Disorder” tended to mirror their internal stabilities, with the former being predicted with high internal consistency and high external predictivity, while both measures were significantly lower for the latter. This trend is further found in Figure 2 via an associative trend (R = 0.62, with a Neyman 95% confidence interval of {0.32, 0.89}) between internal consistency among genAI replicates and their capacity to predict real-world observations.

### 3.2. Comparisons with Clinical Classification

Table 3 reports the agreement between the total genAI ACR score (the sum of positive criteria for a patient) and clinical SLE classifications for the pre-SC and post-SC patient sets. In this table, one notes a moderate balance between those with a positive SLE determination (45) versus those with negative or inconclusive determinations (33).

Pre- and post-classification patient subsets produced comparable predictive success rates for each set (0.72 and 0.79, respectively), yielding a net predictive accuracy of 76% (95% C.I. {71%,81%}). Note that post-SC specificity could not be computed due to the absence of clinical SLE negatives (i.e., 0 “True Negatives”). It is noted that genAI analysis correctly categorizes four of these six true positives for the pre-SC set but produces nine false positives, leading to a poor positive predictive value. This may prompt caution in using the profiler to predict the SLE status of patients without prior classification, although the method still correctly classifies most SLE-negative patients.

## 4. Discussion

The availability and efficacy of medical treatments have advanced at a tremendous rate in recent generations [52], even for therapeutically challenging autoimmune conditions like SLE [19,53,54]. Unfortunately, a greater treatment arsenal does not necessarily equate to commensurate improvement in health outcomes [55], since the number and efficacy of available treatments do not mean that the right patients will receive the right treatments [55]. Rather, a frequent impediment arises from imperfect understanding of patient medical history, often exacerbated by informational challenges such as parsing records of disparate format [56,57] or excessive volume [58].

While AI has helped to spur many valuable recent developments in medical technology, it has not yet been embraced as a means for overcoming these informational impediments. GenAI offers to remedy such challenges through the automated assimilation of medical details contained in these complex, voluminous case histories, but skepticism has persisted regarding its analytical reliability [59,60,61]. The scope of this proof-of-concept study, therefore, is not to present a polished solution claiming highly accurate recapitulation of medical files but rather to assess current strengths and weaknesses of such a proposition. Presenting the record profiling concept in conjunction with realistic performance assessments for a complex disease such as SLE may hopefully provide the basis for continued refinements to the point of delivering practical benefit to patients and their rheumatologists.

One value measure for such a genAI tool is accuracy in informing physicians and their patients regarding medically known factors (i.e., prior medical observations or test results), or combinations thereof, that might identify the risk of a given disease state. Disease-relevant factors have been investigated extensively for classifying diseases such as SLE [9,10,11,12,13,14,15,16,17,18,19,20,21], but definitive SLE classification remains challenging as evinced by findings that the average time to reaching classification is well over seven years, wherein such delays may lead to care fragmentation associated with adverse outcomes [5,18,62]. Conversely, accelerated SLE classification has been shown to lead to fewer flares, fewer instances of hospitalization and a reduced risk of severe organ damage [18,63]. Given such guidance, our study has sought to develop genAI strategies to pinpoint information of possible relevance to tasks like classification. To this end, we have reported a novel genAI tool for assimilating an extensive case history and flagging observations or measurements that may shed light on SLE-relevant determinants, as defined by the ACR 1997 SLE classification protocol. To date, our literature reviews have not identified other development efforts with this informational orientation, and the list of all FDA-approved “Software as Medical Devices” (SaMD) tools does not include any devices oriented toward medical record analysis, generative AI or rheumatology [64].

As an initial demonstration of targeted genAI-based medical profiling, the present study chose a computational replication of the ACR 1997 SLE classification criteria as a representation of medically relevant observables reflecting complex and challenging case heterogeneity. Since the ACR criteria rely on both descriptive and quantitative observables, this choice aligns with the document assimilation strengths of LLM-based genAI. As further motivation, the ACR protocol remains actively used in rheumatological practice. According to a recent meta-analysis, the ACR 1997 criteria achieve SLE classification with a strong specificity of 0.922 (95% Neyman C.I. = {0.871,0.955}) [47], despite requiring fewer distinct determinations than other popular standards such as the 2019 ACR/EULAR [10] and SLICC [11]. Furthermore, the accurate determination of a minor subset of ACR 1997 criteria (i.e., as few as four positive criteria) may produce an SLE classification with high specificity.

Within a limited set of 27 patients for whom full clinical criterial determinations were made, all eleven criteria were correctly predicted more than half the time, albeit such performance was only statistically significant for eight criteria, for which clinical corroboration rates ranged from 0.70 to 1.00. The remaining three criteria (“10. Immunologic Disorder” (0.56), “5. Non-Erosive Arthritis” (0.59) and “4. Oral Ulcers” (0.63)) did not attain statistically significant net predictivity.

Notably, the two criteria best reproduced by genAI, “2. Discoid Rash” and “6. Pleuritis or Pericarditis” both relate to genAI determinations whose clinical and genAI determinations were all “negative”; i.e., final assessments by clinicians consistently found no cases that would support such determinations, and genAI independently corroborated all of these negations. This is a favorable validation of the model’s conservative specifications (i.e., using a low generative temperature of 0.1, and reliance on precise CAG-augmented prompting) as a means for reducing genAI hallucinations that could yield medically problematic false positives. However, a converse implication of using a low-temperature CAG scheme is reflected in the weaker genAI determination of “10. Immunologic Disorder”, whereby poor specificity (all errors are false positives) arises from the challenge of tackling a semantically complex criterion without adequate training over the full range of acceptable positive determinations. For illustrative purposes, the following evidence string from Appendix A (dates are rendered as XXXX; all other details are transcribed from the medical record without modification) produced a false positive determination:



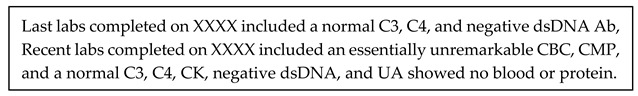



This statement conveys an extensive list of prospective immunopathological markers, all with negative indications, thus suggesting a logically negative verdict. Generative adversarial network analysis, however, is trained not for logic but for pattern scoring. To this end, a general purpose LLM with only modest rheumatological training might see the long list of markers and infer similarity to an immunologic-positive case, especially compared to other patients not being monitored for autoimmune conditions. Furthermore, tendencies toward erroneous determination may also arise from the phrase “a normal C3, C4” (immunologic-negative evidence), which is typographically nearly identical to the immunologic-positive phrase, “abnormal C3, C4.”

Such systematic deficiencies arise from the fundamentally stochastic nature of generative adversarial networks. This stochastic representation is both a strength and a weakness. Such imprecision is thus likely to remain a core limitation of genAI-based medical record profiling. Fortunately, the risks of similarity-based misinterpretation of medical evidence may be reduced by CAG or RAG strategies; i.e., RAG augmentation by a broad representation of varying medical phraseology, or selective CAG infusions intended to encode the most common representations, might better capture the logical implications of different phrasing modes and their relationship in discriminating clinical positives and negatives.

Diagnostic clarity for criterion 5, by contrast, is obfuscated by the challenges of differentiating non-erosive joint problems from manifestations in rheumatoid arthritis [41,65], by significant patient-to-patient variation in symptom presentation [66] and by the complex nature of making definitive clinical characterizations, even with imaging techniques [41] which are not available for text-focused genAI assessments. Finally, criterion 4 (“Oral Ulcers”) poses analytical challenges due to substantial variability in location (most commonly palate, but also cheeks, gums, lips, tongue and throat), appearance (red, white or red-to-white, small to large, raised or depressed), and pain level (ACR instructions are often interpreted as stipulating minimal pain, but outliers are fairly common) [67,68]. A further complication in assessing oral ulcers as a specific symptom of SLE arises in differentiation relative to infections, physical injuries, stress or dietary imbalances [69,70].

Fortunately, while specific examples such as ACR criteria 4, 5 and 10 are illustrative of the diagnostic challenges shared by clinicians and genAI alike, the ACR 1997 standard achieves reliability by balancing the determination across the eleven criteria described in Table 1, and incorporating flexibility such that a positive clinical SLE determination can be meaningfully made if as few as four criteria are met. Within this scoring framework, the composite genAI assessment produces a statistically significant classification accuracy level of 0.76 (95% Neyman C.I.: {0.71,0.81}) relative to clinical rheumatologists’ determinations. These predictive accuracy levels do compare favorably with those in a recent broad-spectrum meta-analysis conducted by Takita et al. [71] that surveyed 54 recent publications attempting genAI strategies for diagnostic determinations across 17 different medical disciplines. A meaningful head-to-head performance comparison is not possible due to methodological differences (i.e., the meta-analysis included both real-world medical diagnoses and hypothetical instructional vignettes) and the use of now-outdated LLMs (surveyed publications dated from 2018 to 2023), but it is nonetheless worth noting that the net concordance across all 54 papers was only 0.569 (95% C.I.: {0.510,0.627}) [71]. Interestingly, although the Takita study did not explicitly target SLE classifications, it did consider undifferentiated diagnoses for 132 “inflammatory rheumatic disease” (IRD) cases [72]. It may be further noted that genAI diagnostic performance on the focused IRD subset (conducted using a 2023 version of ChatGPT4) was weaker than on the superset, achieving a concordance of only 35% [72].

As with any pilot study such as this, success is defined by a balance between an assessment of initial strengths and weaknesses and projection of realistic opportunities for further refinement. In this context, it is fair to note that the tool in our study and most of the other tools reported in the aforementioned Takita meta-analysis [71] are of limited medical impact, since genAI predictivity generally falls significantly short of clinical diagnostic/classification standards, including for complex disorders such as SLE [9,10,11]. Fortunately, there are various avenues for incremental performance improvements to occur. Evidence that competent genAI predictions from 2023 are apparently being outperformed by 2024 LLMs raises the question of what the eventual accuracy ceiling will be for genAI assimilation of medical information, and to what extent LLM customization (e.g., language models specifically trained on medical records, case reports and other diagnostically relevant texts), expert algorithmic tweaks (i.e., successors to current generative adversarial networks and variational autoencoders) and carefully designed LLM prompts may push the limits for the medically reliable assimilation of patient records.

In terms of furthering the future landscape available for genAI support of medical endeavors, it is also important to recognize progressive refinements in the systematization of medical criteria that are available for automated case profiling. In this sense, whereas ACR 1997 served as an instructive initial choice for demonstrating how genAI methods might complement conventional medical determination of symptoms and pathological features, it is only a snapshot in the refinement of disease characterization and classification protocols. Other classification protocols already provide better sensitivity for full SLE classification without loss of specificity. In particular, the newer (and more complex and quantitative) 2019 EULAR/ACR protocol has superseded ACR 1997 as the most widely used protocol among American rheumatology practices, courtesy of exceptionally high rates for both sensitivity (0.961) (a significant improvement over the ACR 1997 sensitivity of 0.842 (95% C.I. {0.762,0.899}) [50]) and specificity (0.934) [10]. The Systemic Lupus International Collaborating Clinics (SLICC) classification [11] also deserves attention as a classification protocol with better sensitivity than ACR 1997, especially for childhood-onset SLE [73,74].

From an applications perspective, there is likely to be value in computationally profiling cases according to conceptually different foci, such as SLE disease activity tracking and flare risk assessment. Examples of activity-related and flare-related frameworks include the Systemic Lupus Erythematosus Disease Activity Index 2000 (SLEDAI-2K) [20], plus flare-oriented measures such as the BILAG-2004 index [21], the SELENA-SLEDAI flare index [22] and the revised SELENA flare index [23], as well as an algorithm-based flare risk index (FRI) laboratory testing using a combination of 11 plasma mediators to predict patient risks for clinical flares in the ensuing three months [75].

Finally, it is valid to assert that the emergence of more universal and standardized electronic health record (EHR) services may facilitate access to convenient search and tabulation capabilities that may diminish the impact of genAI-based record contextualization. Nonetheless, the practicable utilization of EHRs has been implicated in an information overload phenomenon with documented repercussions for both safety and efficacy [59], which again favors the use of genAI assimilation as an adjuvant technology.

While the technological and procedural advances identified in the prior four paragraphs all point toward prevailing opportunities for genAI-based profiling, we should make the caveat that prior segments identify technical deficiencies that must be overcome before asserting a viable contribution to medicine. This is true both for the case profiler concept being proposed in this paper and for the numerous other genAI diagnostic tools surveyed in the Takita meta-analysis [71].

The perception that genAI tools face distinct implementational barriers is accentuated by a current absence (ca. 10/2024) of genAI medical record tools within the ranks of FDA-approved SaMD devices [64]. This perception is furthered by a stipulation in Figure 1 of the “8 December 2017 Guidance to Industry and Food and Drug Administration Staff” that requires software-based devices to analytically and clinically prove an ability to produce “*accurate, reliable and precise*” outputs [64]. While accuracy and reliability are quantifiable and amenable to statistical improvement by tailoring LLM training material and algorithmic adjustments, the term “precise” clashes with genAI due to the stochastic underlying nature of its determinations. Specifically, a well-tuned stochastic procedure may mitigate avoidable errors down to statistically acceptable levels, but may also yield rare erratic (i.e., unpredictable) results that hinder the reliable estimation of “precision”. This could hinder a patient’s understanding of the risks of receiving illogical (and potentially harmful) guidance. To this end, validation of genAI-based medical record profiling tools may require attention not just to standard statistical performance metrics, but also to specialized assessment of rare event risk, such as might be reasonably quantified via measures such as the Rare Event Concentration Coefficient [76].

Additional hurdles, not just for genAI applications but equally for other computational medical record analyses, revolve around data security and privacy. All such efforts must ensure (1) the health privacy for any patients whose existing medical records have been used to illuminate or refine (via LLM training or CAG) computed determinations or predictions, and (2) the prevention of leaks of sensitive information associated with any use of the service.

Avoiding privacy risks to human subjects on whom models are based is facilitated by the responsible use of de-identified medical record data, of which there are now numerous distinct registries implementing rigorous privacy standards. Such registries include those affiliated with disease research foundations, clinical institutions (hospitals, research centers, etc.), government sources such as the Joint Pathology Center https://www.jpc.capmed.mil/ (accessed on 7 February 2025) and HealthData.gov https://healthdata.gov/ (accessed on 7 February 2025) and shared public repositories, such as Synapse https://www.synapse.org/ (accessed on 7 February 2025) and Vivli https://vivli.org/ (accessed on 7 February 2025).

The second assurance poses an evolving challenge, due ongoing competition between protocols aiming to ensure secure data storage and transmission versus malicious exploits to circumvent such security [77,78]. The continually adapting protocols suggest that many security standards written now are likely to soon be outdated. However, several key precautions may sustainably reduce the risk of failure, including the following:Devise a centralized service that does not store a user’s health data for any longer than is required for genAI processing to service the user’s immediate request;User queries to the centralized genAI processor should be initiated with a downloadable app that reads, de-identifies and encrypts all patient data on the client end so that no un-encrypted, identifiable information is communicated over public networks;Ensure that client/server encryption keys are protected by ensuring that only the app can read the client-side key, and by arranging for client/server key pairs to be altered frequently, ideally just prior to initiating any new data transfer.

Having summarized both an array of promising opportunities and sobering challenges, we anticipate that the articulated balance will offer practical guidance for further refinement. The results obtained from this pilot study should also provide an objective snapshot of the potential value of genAI as a means for overcoming informational barriers in distilling medically actionable insight from medical records. If facts and context are twin pillars of knowledge, then advancements such as this should help the pursuit of improved medical outcomes for generations to come.

## Figures and Tables

**Figure 1 biotech-14-00015-f001:**
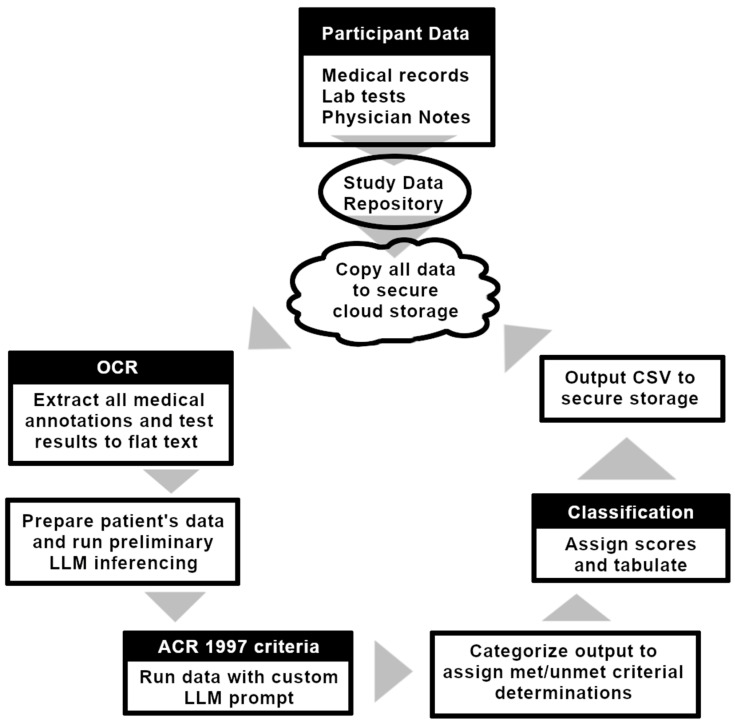
Workflow for LLM-based interpretation and analysis of patient medical records.

**Figure 2 biotech-14-00015-f002:**
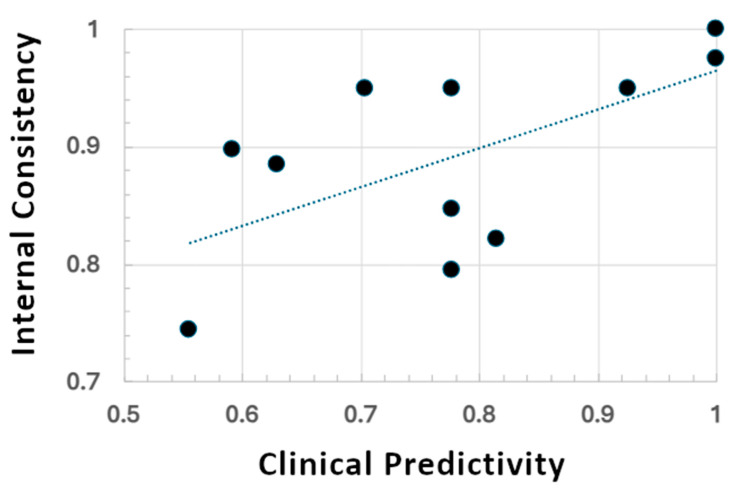
Comparison of internal consistency (*y*-axis) for genAI-predicted ACR 1997 criteria vs. concordance with clinical determinations (*x*-axis). The dotted line is a correlation fit.

**Table 1 biotech-14-00015-t001:** ACR criteria and usage notes. Criteria are weighted equally; positive SLE characterization is made for patients who are flagged as positive in four or more of these eleven criteria.

Criteria	Key Points of Determination
**1. Malar Rash**	Abnormal redness in cheeks, generally absent on nose or near lips [37].
**2. Discoid Rash**	Raised red, scaly patches on sun-exposed skin, rough shape/size of coins; may present follicular plugging; possible long-term scarring [38].
**3. Photosensitivity**	Visual rash, itches, pain or lesions after sun-exposure; should be distinguished from non-SLE disorders (e.g., eczema or dermatomyositis) or sensitivities from medications or allergies; may require photoprovocation [39].
**4. Oral Ulcers**	Visual red sores or red with white halo; usually painless; should distinguish from lichen planus, oral candidiasis, dietary deficiencies or medication side-effects [40].
**5. Nonerosive Arthritis**	Pain, swelling and possible effusion in 2+ peripheral joints; involves synovial dysregulation rather active than tissue degeneration; may require MRI, ultrasound or X-ray [41].
**6. Pleuritis or Pericarditis**	Pleuritic pain, plus pleural rub or pericardial friction rub sounds (via stethoscope); may require chest X-ray or electrocardiogram [42].
**7. Renal Disorder**	Cell casts (red, hemoglobin, tubular and/or granular cells), or test for albumin/creatinine ratio > 500 mg/g [43], Complement C3 < 60 mg/dL and C4 < 15 mg/dL [44], anti-dsDNA titers ≥ 3 [44]; possible blood urea nitrogen [45].
**8. Neurologic Disorder**	Seizures or psychosis; assess strength, coordination, reflexes, neurocognition, and history of sensory anomalies; rule out medication effects, uremia, ketoacidosis and electrolyte imbalance; may entail electroencephalography [46].
**9. Hematologic Disorder**	Tested via Complete Blood Count, simultaneously assessing deficiencies in red blood cells (anemia < 4 million cells/mcL), white (leukopenia < 4000 cells/mcL, recurring) and platelets (thrombocytopenia < 1500 cells/mcL) [47].
**10. Immunologic Disorder**	Recent incidence of fatigue and fever, elevated anti-Sm readings (> 7 u/mL) [48], anti-dnDNA > 15 u/mL, plus other acceptable testing indications for the presence of phospholipid antibodies [49].
**11. Antinuclear Antibody**	Positive antinuclear antibody (ANA); threshold unspecified in 1997 standard, but common positive standard has been a titer of at least 1:80 [50].

**Table 2 biotech-14-00015-t002:** genAI predictions of ACR criteria for 27 patients with fully documented ACR classification records.

	AI Internal Consistency (*)			External Predictivity (#)
ACR Criteria	High	Medium	Low	(Clinical Agreement) ($)
**1. Malar Rash**	0.95	0.00	0.05	0.78 (*p* < 0.003)
**2. Discoid Rash**	1.00	0.00	0.00	1.00 (*p* < 10^−6^)
**3. Photosensitivity**	0.95	0.01	0.04	0.70 (*p* < 0.03)
**4. Oral Ulcers**	0.88	0.09	0.01	0.63 (*p* < 0.12)
**5. Nonerosive Arthritis**	0.90	0.04	0.06	0.59 (*p* < 0.22)
**6. Pleuritis or Pericarditis**	0.97	0.01	0.01	1.00 (*p* < 10^−6^)
**7. Renal Disorder**	0.85	0.10	0.04	0.78 (*p* < 0.003)
**8. Neurologic Disorder**	0.95	0.03	0.03	0.93 (*p* < 0.0003)
**9. Hematologic Disorder**	0.79	0.14	0.06	0.78 (*p* < 0.003)
**10. Immunologic Disorder**	0.74	0.17	0.09	0.56 (*p* < 0.35)
**11. Antinuclear Antibody**	0.82	0.10	0.08	0.81 (*p* < 0.001)

**Notes:** (*) Internal consistency reflects the fraction of 78 patients for whom 5-fold sampling sets of predicted criteria exhibited perfect agreement, versus moderate (single dissent) and low (two dissenting) consistencies. (#) External predictivity reflects the rates for successful AI prediction of clinical criterial determinations. ($) As determined from Fisher’s Exact Test, the *p*-significance values represent the null hypothesis relative to predictivity above random levels.

**Table 3 biotech-14-00015-t003:** Predictive performance for generative AI ACR classification of 78 patients (39 from the post-SC group; 39 from pre-SC). Entries labeled “n/a” have denominators of value 0.

	**Clinical Positives**	**Clinical Negatives**	**Correct Positives**	**Predictive Accuracy**
**Full Set**	45	33	35	0.76
**Pre-SC**	6	33	4	0.72
**Post-SC**	39	0	31	0.79
	**Positive** **Pred. Value**	**Negative** **Pred. Value**	**Sensitivity**	**Specificity**
**Full Set**	0.80	0.71	0.78	0.73
**Pre-SC**	0.31	0.92	0.67	0.73
**Post-SC**	1.00	n/a	0.79	n/a
	**True Pos**	**False Pos**	**False Neg**	**True Neg**
**Full Set**	35	9	10	24
**Pre-SC**	4	9	2	24
**Post-SC**	31	0	8	0

## Data Availability

The original contributions presented in this study are included in the article/Appendix A. Further inquiries can be directed to the corresponding author.

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
