# Peer review of "Prediction of Lupus Classification Criteria via Generative AI Medical Record Profiling"

_biotech, 2025, doi:10.3390/biotech14010015_

Round 1
Reviewer 1 Report
Comments and Suggestions for Authors
The paper is well written and the findings are useful.
What is the ACR?
Authors may strengthen introduction with more references and literature review on lupus etc., it diagnosis, etc.
Include discussion of scope and limitations.
More on validation, perhaps, RAG?
Author Response
Please refer to attachment for full response.

Reviewer 2 Report
Comments and Suggestions for Authors
Dear authors, Your work presents an initial analysis of the use of generative AI in the classification of systemic lupus erythematosus (SLE), exploring how these models can aid in diagnosis and clinical management. The topic is timely and relevant, and the article describes the application of language models to specific clinical criteria effectively.
However, I believe the introduction could be enhanced by incorporating information on the clinical and economic impact of SLE to better contextualize the study’s importance, as well as by clearly distinguishing what makes this proposed approach innovative.
The results section would benefit from a deeper analysis, particularly of the criteria that performed below expectations, and more quantitative data on the statistical tests conducted should be included.
I also recommend that the discussion delve further into the ethical and implementation challenges of using AI in clinical environments, addressing privacy concerns and comparing the findings with results from other studies utilizing AI in medical diagnostics.
I believe these suggestions could improve the clarity and rigor of the work.
Author Response

(The authors gave the same response as above.)

Round 2
Reviewer 2 Report
Comments and Suggestions for Authors
Dear Authors, congratulations on the work developed; the article seems clearer and more rigorous to me.